# Usefulness of Body Position Change during Local Ablation Therapies for the High-Risk Location Hepatocellular Carcinoma

**DOI:** 10.3390/cancers16051036

**Published:** 2024-03-03

**Authors:** Hitomi Takada, Yasuyuki Komiyama, Leona Osawa, Masaru Muraoka, Yuichiro Suzuki, Mitsuaki Sato, Shoji Kobayashi, Takashi Yoshida, Shinichi Takano, Shinya Maekawa, Nobuyuki Enomoto

**Affiliations:** Gastroenterology and Hepatology Department of Internal Medicine, Faculty of Medicine, University of Yamanashi, Yamanashi 409-3898, Japan; ykomiyama@yamanashi.ac.jp (Y.K.); reonao@yamanashi.ac.jp (L.O.); mmuraoka@yamanashi.ac.jp (M.M.); yuichirohs@yamanashi.ac.jp (Y.S.); satom@yamanashi.ac.jp (M.S.); shoji@yamanashi.ac.jp (S.K.); tyoshida@yamanashi.ac.jp (T.Y.); stakano@yamanashi.ac.jp (S.T.); maekawa@yamanashi.ac.jp (S.M.); enomoto@yamanashi.ac.jp (N.E.)

**Keywords:** hepatocellular carcinoma, local ablation therapies, body position change

## Abstract

**Simple Summary:**

Local ablation therapies are important treatment options for hepatocellular carcinoma (HCC). Some HCC nodules were classified as HCC in high-risk locations based on ultrasound (US) if they were adjacent to large vessels or extrahepatic organs or if they were poorly visible on US. Various techniques have been used to perform these therapies efficiently and safely. However, few reports have discussed the usefulness of body position change (BPC). The aim of our study was to assess the usefulness of BPC during radiofrequency ablation therapies in patients with HCC. We confirmed the high technical success rates in patients with BPC, and no differences in the procedure time, local tumor progression rates, intrahepatic distant recurrence rates, and overall survival between the groups with and without BPC in the high-risk location group. Performing BPC during local ablation therapies in patients with HCC in high-risk locations may enable safe and accurate treatment.

**Abstract:**

Local ablation therapies are important treatment options for early-stage hepatocellular carcinoma (HCC). Various techniques have been used to perform these therapies efficiently and safely. However, few reports have discussed the usefulness of body position change (BPC). This study aimed to investigate the usefulness of BPC during local ablation therapies in patients with HCC. We evaluated 283 HCC nodules that underwent local ablation therapy. These nodules were categorized into high- or low-risk locations on the basis of their proximity to large vessels, adjacent extrahepatic organs, or poor visibility under ultrasound (US) guidance. The technical success rates, procedure time, and prognosis were evaluated. In this study, 176 (62%) nodules were classified in the high-risk location group. The high-risk location group was treated with techniques such as BPC, artificial pleural fluid, artificial ascites, fusion imaging, and contrast-enhanced US more frequently than the low-risk location group. The technical success rates were 96% and 95% for the high- and low-risk location groups, respectively. Within the high-risk location group, those without BPC had a lower success rate than those with BPC (91% vs. 99%, *p* = 0.015). Notably, BPC emerged as the sole contributing factor to the technical success rate in the high-risk location group (OR = 10, 95% CI 1.2–86, *p* = 0.034). In contrast, no differences were found in the procedure time, local tumor progression rates, intrahepatic distant recurrence rates, and overall survival between the groups with and without BPC in the high-risk location group. In conclusion, BPC during local ablation therapy in patients with HCC in high-risk locations was safe and efficient. The body position should be adjusted for HCC in high-risk locations to maintain good US visibility and ensure a safe puncture route in patients undergoing local ablation therapies.

## 1. Introduction

Hepatocellular carcinoma (HCC) is the sixth most commonly diagnosed cancer and the fourth leading cause of cancer-related death [1]. The main treatments for HCC include local ablation therapies (such as radiofrequency and microwave ablation (RFA and MWA, respectively)), surgical resection, transarterial chemoembolization, and liver transplantation. Local ablation therapy is particularly beneficial for patients with hepatic decompensation or those who are ineligible for surgical resection because of complications [2]. This therapy has less impact on residual liver function than resection, making it an important treatment option for early-stage HCC [3,4,5,6].

The 3- and 5-year survival rates for patients treated with RFA were 88.9% and 76.6% for liver damage A and 75.2% and 56.5% for liver damage B, respectively. Similarly, the 3- and 5-year survival rates of patients with HCC treated by MWA were 72–73% and 57–60%, respectively [7,8,9]. The SURF trial investigated the effects of surgical resection and RFA in patients with HCC with a maximum diameter of <3 cm and a number of tumors of less than 3. The median recurrence-free survival was 2.98 and 2.76 years in the resection and RFA groups (hazard ratio 0.96, *p* = 0.793), respectively [10]. Furthermore, randomized controlled prospective trials have demonstrated the equivalence of these two therapies. Hence, when performing local ablation therapies, it is important to accurately control HCC equivalent to resection and minimize complications.

Advancements in ultrasound (US) technology and various techniques have enhanced the accuracy and safety of local ablation therapies. These include RFA with contrast-enhanced US (CEUS), real-time virtual sonography (RVS), artificial pleural and ascites fluid infusion, and body position change (BPC). Implementation of these methods has seemingly improved the safety and completion rates of local ablation therapies. The efficacy of RFA along with CEUS, RVS, and artificial fluid infusion has been supported in previous reports [11,12,13,14,15,16,17,18]. However, the benefit of BPC in patients treated with RFA was not adequately investigated. Therefore, this study focused on the usefulness of BPC during local ablation therapy in patients with HCC.

## 2. Materials and Methods

### 2.1. Patients

This study included 283 nodules in 230 patients with HCC who received RFA therapies at our institution between January 2018 and 2022 (Figure 1). The inclusion criteria were as follows: patients (1) >18 years old; (2) with performance status of 0 or 1; (3) who had surgically unresectable HCC lesions or who voluntarily chose nonsurgical treatment; (4) with three or fewer lesions and ≤5 cm in diameter; (5) with total serum bilirubin concentration of <3 mg/dL; and (6) without extrahepatic metastasis or vascular tumor invasion. Patients with (1) a platelet count below 50 × 10^9^/L, (2) prothrombin activity below 50%, (3) refractory ascites, and (4) nodules suspected of poorly differentiated HCC on pretreatment imaging and patients (5) receiving a combination of local and other therapies were excluded from this study [19,20,21]. In cases with multiple nodules, cases with two nodes in close proximity to each other, such that the predicted area of ablation at the time of ablation of target nodule included the other nodule, were excluded.

Nodules that showed pathological examination result, or non-rim hyperenhancement in the arterial phase of dynamic computed tomography (CT) or gadolinium ethoxybenzyl diethylenetriamine penta-acetic acid-contrast-enhanced magnetic resonance imaging (MRI) and non-peripheral washout or threshold growth, that is, only nodules that showed LR-4, 5 using LI-RADS (Liver Imaging Reporting and Data System) were diagnosed as HCC [22]. We defined ‘diameter of the tumors’ as the maximal diameter obtained using axial section CT images at 1-mm intervals, coronal section images at 3-mm intervals or sagittal section images at 3-mm intervals. In this study, patients with hepatitis B virus (HBV), hepatitis C virus (HCV), and non-B non-C hepatitis were defined as those having detectable HBV-DNA or receiving nucleoside analog therapy and those with detectable HCV-RNA or sustained virological response.

The study was conducted in accordance with the Declaration of Helsinki and was approved by the Human Ethics Review Committee of the University of Yamanashi. Written informed consent was obtained from all patients.

### 2.2. RFA Techniques and Devices

Percutaneous local ablation therapy was performed on an inpatient basis. Two experienced operators previewed the targeted nodule using US (planning US) the day before treatment. The US visibility of the target tumor was graded by the operators on a 4-point scale: invisible, poor, fair, and good [23]. Thereafter, the operators assessed the presence of a safe access route for electrode placement.

The devices used for RFA were the Cool-tip RFA system (Covidien, Boulder, CO, USA), and VIVA RF system (STARmed, Goyang-si, Republic of Korea) [24]. After local anesthesia administration, the electrode was inserted under US guidance (Aplio 500, Toshiba, Tokyo, Japan) using a dedicated US transducer for puncture (PVT-350BTP, Toshiba, Japan). In the case of 3-cm internally cooled tip RF electrodes, the output was initiated at 60 W and increased by 20 W/min until tissue impedance overshooting occurred using the manual i mode and watching the output display on the generator. With 2-cm internally cooled tip RF electrodes, the output started at 40 W and then increased by 10 W/min until tissue impedance overshooting occurred. In the case of the adjustable VIVA RF electrode, we used the general mode, automatically monitoring the impedance rise and adjusting the radio wave output.

All ablation procedures were performed by one of three experienced physicians with 12 years of experience in performing US-guided percutaneous ablation.

### 2.3. HCC in High-Risk Locations

HCC nodules were classified as HCC in high-risk locations if they were adjacent to large vessels or extrahepatic organs or if they were poorly visible on US. Nodules were considered adjacent to large vessels if they were located <5 mm from the first or second branch of the portal vein, the base of hepatic veins, or the inferior vena cava, referring to previous reports [5]. Nodules located <5 mm from the diaphragm, heart, lung, gallbladder, right kidney, or gastrointestinal tract were considered adjacent to extrahepatic organs. The distance between the edge of the nodule and the large vessel or extrahepatic organ was measured using axial section CT images reconstructed at 1-mm intervals, coronal section images at 3-mm intervals and sagittal section images at 3-mm intervals.

### 2.4. Treatment-Assist Techniques for HCC in High-Risk Locations

To protect against thermal injury during the procedure, an artificial pleural fluid infusion method with 5% glucose was used if the targeted nodules were close to the diaphragm or lungs. The artificial ascites effusion method was used if the nodules were close to other extrahepatic organs in the abdominal cavity [25]. For patients whose B-mode US could not identify the nodule, we used artificial pleural effusion, artificial ascites, CEUS, RVS, or BPC [26,27,28]. An image analysis system (Volume Analyzer Synapse VINCENT, version 5.1, Fujifilm Medical Systems, Tokyo, Japan) was used to create fusion images [23,29].

For BPC, patients were examined in various positions, including supine, half side lying (right and left), head up, and upright (Figure 2) [30,31]. Patients with no problems with image characterization of HCC using US and a safe puncture route were treated in the supine position. The right half side lying position was used primarily for HCCs in segments S2, 3, and 4, whereas the left half side lying position was used for those in segments S6, 7, and 1. The head up and upright positions were used for those in the S7 and 8 segments, particularly in the liver dome. BPC was attempted not only in these locations, but also in many cases where we want to further improve the image characterization using US, where the nodule is classified as high-risk HCC and we need a safer puncture route, or where are expected to be difficult to puncture with the basic approach due to obesity and postoperative, etc. If BPC was judged to be effective when we performed planning US the day before treatment, all procedures started after appropriate positioning using a soft cushion, arm-board and support device. Treatment-assist techniques in planned combinations were performed as follows; artificial pleural fluid or ascites infusion was performed before BPC for puncture, and RVS and CEUS were performed just before puncture after BPC.

Starting in October 2019, after one hepatologist received training at the high-volume center, we actively used BPC during ablation therapies. Patients were classified into phase 1 (January 2018–December 2019) and phase 2 (January 2020–January 2022) before and after the proactive use of BPC.

### 2.5. Assessment of Treatment Response and Follow-Up

A hepatologist and two radiologists assessed the therapeutic response. Treatment response was assessed using plain CT at 1–3 days after the ablation and dynamic CT at 1 month after the ablation. Later, dynamic CT or EOB-MRI was performed every 3 months. Nodules showing no CT/MRI evidence of residual tumor with continuous monitoring <3 months, i.e., nodules that achieved ‘LR-TR nonviable’ were defined as ‘technical success’ [22]. In cases of suspected recurrence, imaging studies were conducted. Local tumor progression was defined as the appearance of a viable tumor contiguous with the original nodal site during follow-up. All intrahepatic recurrences, excluding local tumor progression, were defined as intrahepatic distant recurrences.

### 2.6. Statistical Analysis

Median and range were used to report continuous data. To analyze group differences, the study used the Mann–Whitney U test, Kruskal–Wallis test, and nonparametric ANOVA. When one-way ANOVA yielded significant outcomes, Fisher’s exact test was used to examine the differences between specific groups. The Kaplan–Meier method was used to determine both progression-free survival and overall survival (OS), with the log-rank test used for analysis. Statistical significance was defined as a *p*-value < 0.05. Statistical analyses were performed using EZR (version 1.64, Saitama Medical Center, Jichi Medical University, Saitama, Japan), a graphical user interface for R (The R Foundation for Statistical Computing, Vienna, Austria) [32].

## 3. Results

### 3.1. Patients

This study involved 283 nodules in 230 patients with a median follow-up period of 52 (45–55) months. Table 1 shows the baseline characteristics of patients receiving RFA therapy. The maximal diameter of the tumors ranged between 4 and 38 mm (median = 12 mm). The number of nodules to be treated was 1 (186 cases), 2 (78 cases), or 3 (19 cases). The most common tumor location was segments 7 and 8 (each 23%). However, 62% of the nodules were classified in the high-risk location group. The following procedures were performed more frequently in the high-risk location group than in the low-risk group: BPC (61% vs. 24%, *p* < 0.001), artificial pleural fluid infusion (24% vs. 5.6%, *p* < 0.001), artificial ascites infusion (50% vs. 11%, *p* < 0.001), fusion imaging (28% vs. 8.4%, *p* < 0.001), and CEUS (26% vs. 4.7%, *p* < 0.001). The use of treatment-assist techniques such as BPC (2.9 vs. 3.6, *p* < 0.001), artificial pleural fluid infusion (2.9 vs. 3.6, *p* < 0.001), artificial ascites infusion (2.8 vs. 3.6, *p* < 0.001), fusion imaging (2.1 vs. 3.3, *p* < 0.001) and CEUS (2.0 vs. 3.5, *p* < 0.001) improved the intraoperative US visibility for high-risk HCC using the four-point scale, compared to the baseline US the day before treatment.

### 3.2. Technical Success of Local Ablation Therapy

The total technical success rate was 96% (167/173 nodules) and 95% (102/107 nodules) for the high- and low-risk location groups, respectively. Notably, within the high-risk location group, nodules treated with BPC achieved a significantly higher technical success rate than those without BPC (99% vs. 91%, *p* = 0.015; Figure 3a). Our analysis identified BPC as the only factor related to the technical success rate for the high-risk location group (OR = 10 (1.2–86), *p* = 0.034; Table 2).

### 3.3. Procedure Time

The treatment duration according to the nodule number is reported in Figure 2b. The high-risk location group in patients with 1 target lesion demonstrated longer treatment duration than the low-risk location group (61 min vs. 43 min, *p* < 0.001). However, no differences were found in the procedure time between the groups with and without BPC in the high-risk location group (*p* = 0.36; Figure 3b).

### 3.4. Local Tumor Progression

The local tumor progression rates for all nodules during the follow-up period were 4.4%, 8.2%, and 12% at 1, 3, and 5 years, respectively (Figure 4a). The local tumor progression rate showed no significant difference between the high- and low-risk location groups (*p* = 0.18; Figure 4b). Similarly, there was no significant difference in the local tumor progression rate between the groups without and with BPC in the high-risk location group (*p* = 0.64; Figure 3c and Figure 4c).

### 3.5. Intrahepatic Distant Recurrence

The intrahepatic distant recurrence rates for all nodules during the follow-up period were 37%, 66%, and 84% at 1, 3, and 5 years, respectively (Figure 4d). The difference in intrahepatic distant recurrence rates between the high- and low-risk location groups was insignificant (*p* = 0.18; Figure 4e). Moreover, there was no significant difference in intrahepatic distant recurrence rates between the groups without and with BPC in the high-risk location group (*p* = 0.65; Figure 3d and Figure 4f).

### 3.6. Overall Survival

The survival rates for all nodules were 97%, 79%, and 58% after 1, 3, and 5 years, respectively (Figure 4g). No significant difference in OS was detected between the high- and low-risk location groups (*p* = 0.16; Figure 4h) or between the groups without and with BPC in the high-risk location group (*p* = 0.83; Figure 3e and Figure 4i).

### 3.7. Complications

The rates of complications were comparable between the high- (4.0%) and low-risk location groups (5.6%, *p* = 0.57). Similarly, they were comparable between the groups with and without BPC in the high-risk location group (2.8% vs. 5.8%, *p* = 0.44).

### 3.8. Association between the Timing of Treatment and Prognosis in the High-Risk Location Group

In the high-risk location group, the technical success rates of phase 1 by one session were lower than those of phase 2 (89% vs. 100%, *p* < 0.001; Figure 3a). The treatment duration of phase 1 (n = 61) was longer than that of phase 2 (n = 115) (83 (12–194) min vs. 63 (15–125) min, *p* < 0.001; Figure 3b). However, the local tumor progression rates, intrahepatic distant recurrence rates, and OS between the phase 1 and 2 groups were comparable (*p* = 0.34, 0.71, 0.34; Figure 3c–e). Compared with phase 1, phase 2 used intraoperative RVS less frequently (18% vs. 48%, *p* < 0.001), whereas BPC was used more frequently (77% vs. 31%, *p* < 0.001) along with artificial ascites infusion (60% vs. 31%, *p* < 0.001; Table 3).

## 4. Discussion

In this study, BPC during local ablation therapies in patients with HCC in high-risk locations was useful. To our knowledge, this is the first study to statistically show that BPC is safe and efficient in patients receiving local ablation therapies. The body position should be adjusted for HCC in high-risk locations to maintain good US visibility and ensure a safe puncture route in patients undergoing local ablation therapies. We believe that BPC should be attempted in all cases where we want to further improve the image characterization using US, or where the nodule is classified as high-risk HCC and a safer puncture route is to be sought. Furthermore, it is important to sufficiently predict the US view before treatment.

The Barcelona Clinic Liver Cancer (BCLC) classification and the consensus-based clinical practice guidelines for HCC management proposed by the Japan Society of Hepatology have been widely used to determine the treatment strategy in Japan [33]. Recurrence-free survival did not differ significantly between the surgery and RFA groups in patients with the largest HCC diameter ≤3 cm and ≤3 HCC nodules (*p* = 0.58) [10]. Therefore, RFA and surgical resection are recommended for patients with BCLC stage A HCC. Especially in today’s aging population, RFA is an important treatment option because it is effective and less invasive. RFA delivers thermal energy via an electric current between the electrode and a ground pad and has been evaluated as a first-line therapy in early HCC. However, a high incidence of local recurrence in approximately 10% of patients, especially when located near large blood vessels, has been reported previously [33]. In contrast, MWA originated in the 1980s and was modified with the improvement of antenna and therapy strategies. It has been the most recent generation ablation system that uses electromagnetic waves that are less prone to the heat sink effect [34]. MWA is as safe and effective as RFA and achieves a high complete ablation rate with the ability to reach a larger target area than RFA [35,36,37]. MWA has an obvious superiority to RFA in larger tumors ≥3 cm in diameter or when it is close to large vessels.

When performing local ablation therapies, it is important to obtain a better sonographic window and plan a safe puncture route. Various body positions have been attempted for better visualization of the entire liver during US examinations [31,38,39]. Among various body positions, supine, right half side lying, left half side lying, head up, and upright were mainly used according to the site of the tumor and the patient’s body size empirically. Although BPC is frequently used during US examinations of the liver, this topic has not been sufficiently explored. Therefore, our study focused on providing statistical evidence of the usefulness of BPC during RFA therapies.

As shown in this study, in the high-risk location group, the technical success rates of the group without BPC were lower than those with BPC. Moreover, only BPC was a factor related to the technical success rate for the high-risk location group. In contrast, no differences were found in the treatment duration between the groups with and without BPC. We believe that the total procedure time is shorter and the rate of complete ablation achieved in one session is higher in the group with BPC, even if it takes more time and effort to perform BPC. In addition, there was no association between BPC and recurrence or survival rates, probably because of the achievement of adequate ablation in multiple sessions, or adequate HCC control with the addition of other therapies. In contrast, in patients with low-risk HCC, the technical success rates in the group with BPC were comparable to those in the group without BPC (100% vs. 94%, *p* = 0.33). There was no difference in the procedure time, local tumor progression rates, intrahepatic distant recurrence rates, and OS between the two groups (*p* = 0.16, 0.28, 0.12, 0.68). Therefore, it is important to have an adequate treatment plan for each nodule respectively before starting local ablation therapies and to ensure a safe and secure puncture route.

Other treatment-assist techniques include artificial pleural effusion, artificial ascites, CEUS, and RVS. Artificial ascites and pleural effusion are frequently used to protect adjacent organs from thermal damage and obtain a better sonographic window. RVS has been used since 2003, and because of improved accuracy in image synchronization, it is used to confirm the position of HCC before puncture and to check the area to be ablated after treatment [23,40]. Recently, US–US overlay fusion guidance for local controllability was reported to be highly effective for safety margin achievement [41]. Moreover, the usefulness of combination techniques such as RVS-guided RFA with artificial ascites has been reported in a few reports [42]. Conversely, CEUS is frequently used for HCC in coarse liver parenchyma using the Kupffer phase [43]. It may also be useful for evaluating treatment response immediately after local therapies in the future, when more patients with poor renal function are probably elderly and have comorbidities and concomitant systemic therapies. In this study, there was no improvement in the success rates of the high-risk location group with these other modalities, although improved intraoperative US visibility for high-risk HCC. It is particularly interesting that the combined RVS was not a factor associated with increased success rates. This may be due to the unplanned use of other modalities, i.e., cases in which intraoperative US visibility was inadequate and a hasty decision was made to use other modalities. RVS was most frequently used in these cases, including those in which RVS was not used effectively. Therefore, it is not possible to conclude from this study whether other modalities such as RVS were useful when they were used in a planned manner, and further study is needed.

One of the three hepatologists conducting local ablation therapies in this study received a practical training program at a high-volume center for 6 months. Ablation therapies from January 2020 onward, incorporating his training experience, have achieved higher control rates, safely, and shorter procedure times compared to before. Undergoing an intensive training program on local ablation therapies is an extremely valuable experience. Although it is difficult to accurately assess the outcomes of the training program at a high-volume center, it is an important opportunity for non-high-volume centers to improve their competence, as our institution experienced in this study. Therefore, the good results of this study are certainly not solely due to the aggressive addition of BPC, and further studies are needed.

In modern treatment for HCC, where immune checkpoint inhibitors for unresectable HCC have been added as a new treatment option, treatment planning for each individual HCC nodule, even in patients with advanced HCC, may lead to improved prognosis rather than broadly classifying the stages. Moreover, RFA can activate tumor-specific T lymphocytes and enhance the killing ability of natural killer cells toward hepatoma cells [44,45]. Local ablation therapies are no longer just for early-stage HCC but will now be required to be used in combination with other therapies in advanced HCC. Accordingly, the establishment of treatment-assist techniques to achieve safer and more accurate complete ablation is essential.

This study had several limitations. First, this is a single-center study with a few cases. Second, bias may exist in patient selection. Most nodules suspected of having poorly differentiated HCC on pretreatment imaging are selected for surgical resection; however, there are a few cases in which the imaging and pathological diagnoses diverge. Additionally, most of the nodules have not undergone tumor biopsy and may have contained high-grade nodules such as poorly differentiated HCC. Third, there is a lack of detailed analysis of the shape of the tumor. Even nodules with the same maximum tumor diameter may differ significantly in their therapeutic efficacy and prognosis, depending on their three-dimensional structure. Fouth, the study only included patients who received RFA therapies and did not include those with MWA, which is widely used these days. There will be a difference in results between patients with RFA and MWA, especially when treating nodules near large vessels, and further studies are needed. Fifth, as noted in Section 4, this study is not considered a result solely of the effective use of BPC. Moreover, there are few cases in which each treatment-assist technique is used alone, and the appropriate technique varies from nodule to nodule. Therefore, it is difficult to prove the advantage of BPC over the other treatment-assist techniques only in this study. However, we believe that BPC is certainly useful during local ablation therapies for high-risk locations of HCC, even when various hepatologists with varying years of experience and competence are involved.

## 5. Conclusions

To the best of our knowledge, this study is the first to statistically propose that BPC is safe and efficient in patients receiving local ablation therapies. The use of treatment-assist techniques such as BPC, artificial pleural fluid infusion, artificial ascites infusion, fusion imaging and CEUS, improved the intraoperative US visibility for high-risk HCC, compared to the baseline US. In particular, within the high-risk location group, nodules treated with BPC achieved a significantly higher technical success rate than those without BPC (99% vs. 91%, *p* = 0.015). BPC was the only factor related to the technical success rate for the high-risk location group (OR = 10, *p* = 0.034). In addition, no differences were found in the procedure time between the groups with and without BPC in the high-risk location group. The appropriate use of BPC when performing local ablation therapies may allow for more precise treatment in a shorter time and safely.

## Figures and Tables

**Figure 1 cancers-16-01036-f001:**
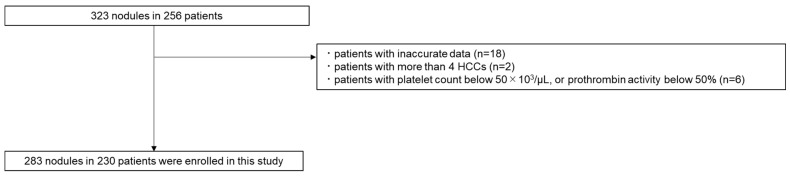
Flowchart of patient enrollment.

**Figure 2 cancers-16-01036-f002:**
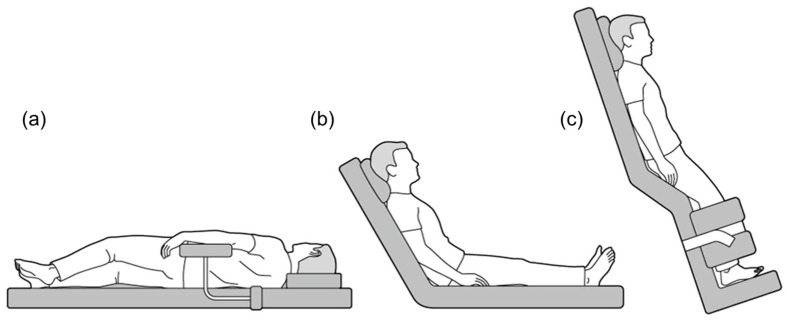
Various positional changes used during local ablation therapies. (**a**) Half side lying, (**b**) Head up, and (**c**) Upright position.

**Figure 3 cancers-16-01036-f003:**
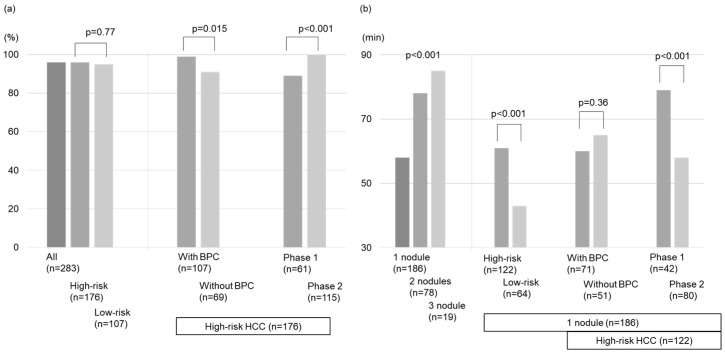
Comparison of outcomes after local ablation therapies. (**a**) The technical success rates, (**b**) The treatment time, (**c**) The local tumor progression rates, (**d**) The intrahepatic distant recurrence rates, and (**e**) Overall survival rates.

**Figure 4 cancers-16-01036-f004:**
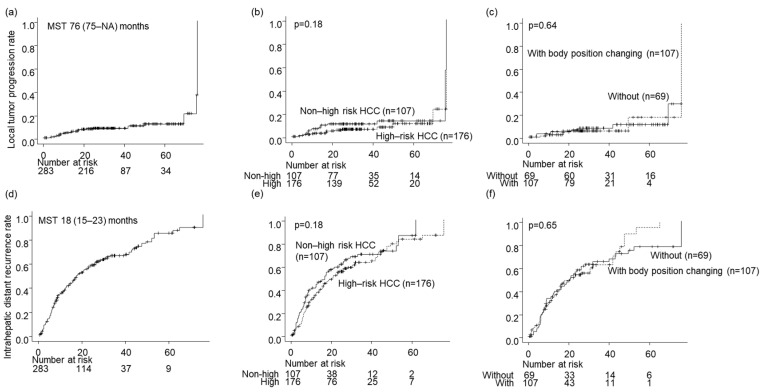
Prognosis after local ablation therapies. (**a**–**c**) The local tumor progression rates, (**d**–**f**) The intrahepatic distant recurrence rates, (**g**–**i**) The overall survival rates. (**a**,**d**,**g**) In all the patients, (**b**,**e**,**h**) By High-risk HCC or not, (**c**,**f**,**i**) with or without body position changing in High-risk HCC.

**Table 1 cancers-16-01036-t001:** Baseline characteristics.

Variables	All	High-Risk Location Group	Low-Risk Location Group	*p* Value
	283 nodules (in 230 patients)	176 nodules	107 nodules	
Age (years)	76 (56–91)			
Male	161 (70%)			
Etiology, HBV/HCV/nonBnonC	18/127/85 (8/55/37%)			
Child-Pugh score, 5/6/7/8/9	192/20/10/4/4 (84/8.7/4.3/2/2%)			
Platelet < 150 × 10^3^/μL)	144 (63%)			
AFP (ng/mL)	4.2 (0.80–2657)			
DCP (mAU/mL)	23 (8.0–4721)			
Primary/recurrence HCC	45/185 (20/80%)			
Maximal diameter of the tumors (mm)	12 (4–38)	12 (4–37)	12 (5–38)	0.32
Number of nodules, 1/2/3	186/78/19	122/46/8 (69/26/5%)	64/32/11 (60/30/10%)	0.15
Tumor location, segment 1/2/3/4/5/6/7/8	2/15/23/35/31/46/65/65 (0.7/5.3/8/12/11/16/23/23%)	1/10/16/22/19/22/40/45 (0.6/5.7/9.1/13/11/13/23/26%)	1/5/7/13/12/24/25/20 (0.6/5.7/9.1/13/11/13/23/26%)	NA
Baseline US visibility scale, 1/2/3/4	15/52/74/142 (5.3/18/26/50%)	15/52/59/50 (8/30/34/28%)	0/0/15/92 (0/0/14/86%)	<0.001
Intraoperative US visibility scale, 1/2/3/4	0/4/74/205 (0/1.4/26/72%)	0/4/65/107 (0/2/37/61%)	0/0/9/98 (0/0/8/92%)	<0.001
Treatment-assist techniques	133 (47%)	107 (61%)	26 (24%)	<0.001
Body position changing				
Artificial pleural fluid infusion	48 (17%)	42 (24%)	6 (5.6%)	<0.001
Artificial ascites infusion	100 (35%)	88 (50%)	12 (11%)	<0.001
Fusion imaging	59 (21%)	50 (28%)	9 (8.4%)	<0.001
Contrast enhanced ultrasonography	51 (18%)	46 (26%)	5 (4.7%)	<0.001

HCC: hepatocellular carcinoma, HBV: hepatitis B virus, HCV: hepatitis C virus, AFP: alpha-fetoprotein, DCP, des-gamma-carboxy prothrombin, US: ultrasound.

**Table 2 cancers-16-01036-t002:** Related factors for the technical success rate for high-risk locations HCC.

Variables	Odds Ratio	Range	*p* Value
Age (years)	0.96	0.86–1.1	0.52
Male	0.5	0.057–4.4	0.53
Etiology, virus	1.1	0.25–5.3	0.87
Child-Pugh score	1.03	0.41–2.6	0.95
Platelet (×103/μL)	1	0.99–1.01	0.99
AFP (ng/mL)	1.01	0.99–1.1	0.86
DCP (mAU/mL)	1.04	0.97–1.1	0.29
Primary HCC	1330	0-Inf	0.99
Maximal diameter of the tumors (mm)	1.1	0.91–1.3	0.31
Number of nodules	0.82	0.25–2.7	0.75
Tumor location, left lobe	0.96	0.18–5.1	0.96
Tumor location, proximity to adjacent extrahepatic organs	0.99	0.21–4.5	0.99
Tumor location, proximity to large vessels	5330	0-Inf	0.99
Baseline US visibility scale	0.96	0.43–2.1	0.91
Intraoperative US visibility scale	0.59	0.12–2.9	0.52
Output power (W)	0.98	0.90–1.1	0.63
Treatment-assist techniques Body position changing	10	1.2–86	0.034
Artificial pleural fluid infusion	0.78	0.15–4.2	0.77
Artificial ascites infusion	1.4	0.29–6.2	0.7
Fusion imaging	0.51	0.11–2.4	0.4
Contrast enhanced ultrasonography	2.2	0.26–19	0.48

HCC: hepatocellular carcinoma, AFP: alpha-fetoprotein, DCP, des-gamma-carboxy prothrombin, US: ultrasound.

**Table 3 cancers-16-01036-t003:** Patient characteristics by the timing of treatment.

Variables	Phase 1 Group	Phase 2 Group	*p* Value
	100 nodules	183 nodules	
Maximal diameter of the tumors (mm)	12 (6–38)	12 (4–37)	0.5
Number of nodules, 1/2/3	67/30/3 (67/30/3%)	119/48/16 (65/26/9%)	0.31
Tumor location, segment 1/2/3/4/5/6/7/8	0/2/2/6/8/9/18/15 (0/2/2/6/8/9/18/15%)	1/8/14/16/11/13/22/30 (0.9/7.0/12/14/10/11/19/26%)	0.30
AFP (ng/mL)	4.4 (1.1–454)	4.4 (0.80–2657)	0.46
DCP (mAU/mL)	24 (10–4721)	23 (8.0–1519)	0.96
Baseline US visibility scale, 1/2/3/4	2/26/30/42 (2/26/30/42%)	13/26/44/100 (7/14/14/55%)	0.012
Intraoperative US visibility scale, 1/2/3/4	0/1/30/69 (0/1/30/69%)	0/3/44/136 (2/24/74%)	0.55
High-risk HCC	61 (61%)	115 (63%)	0.8
Treatment-assist techniques			
Body position changing	21 (21%)	112 (61%)	<0.001
Artificial pleural fluid infusion	19 (19%)	29 (16%)	0.51
Artificial ascites infusion	21 (21%)	79 (43%)	<0.001
Fusion imaging	37 (37%)	22 (12%)	<0.001
Contrast enhanced ultrasonography	15 (15%)	36 (20%)	0.42

AFP: alpha-fetoprotein, DCP, des-gamma-carboxy prothrombin, US: ultrasound, HCC: hepatocellular carcinoma.

## Data Availability

The data presented in this study are available on request from the corresponding author.

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
