# Peer review of "Usefulness of Body Position Change during Local Ablation Therapies for the High-Risk Location Hepatocellular Carcinoma"

_cancers, 2024, doi:10.3390/cancers16051036_

Round 1
Reviewer 1 Report
Comments and Suggestions for Authors
In this contribution, an analysis about how the heat sink effect from large vessels affects the radiofrequency ablation of hepatocellular carcinoma is carried out, with the objective of appreciate if a body position change would improve the ablation procedure. The authors analyze 283 nodules whereas 176 have been classified as high-risk. After presenting how did they choose their samples, technical success rate, procedure time, and prognosis, have been evaluated. A high technical success rate with well-established techniques for ablation has been obtained, with body position change as a relevant actor for this high success rate. The authors conclude at the end that the body position change has a role when hepatocellular carcinoma should be treated in zones where the heat sink effect is important
The topic here investigated is of interest, since it is widely known that a heat sink effect would alterate the HCC ablation procedure. The solution here shown is something that would be helpful for people who work in this area. It is then suggested to consider the present paper for publication after addressing the following points
· When choosing the patients, how did the authors be sure that among their patients they have many of this with lesions close to large vessels? Did they somehow quantify the distance between a lesion and a large vessel, and use a threshold criterion for this?
· Did the authors account for the fact that if lesions are too close then the heat conducted within tissues would affect lesions? Is it fair to include patients with multiple lesions?
· Why did the authors use the Emprint ablation system? Is there an effect on ablation due to the internally cooled electrode, that might depend on the antenna employed?
· The procedure used for ablation might affect results. The authors report in subsection 2.2 that they increase the output power up to 10 W/min until there is a too much high impedance. How did the authors monitor this?
· In this paper, the authors essentially study how a heat sink effect from large vessels might affect thermal ablation. If this is the aim of the present paper, it is suggested to also include in the state of art papers that present a throughout analysis of this effect (doi.org/10.3390/bioengineering10091057). This would underline why investigating this aspect is of primary importance in cancer ablation
· When comparing outcomes from different patients, why didn't the authors give more emphasis to the input power as a variable? This has to compensate the heat sink effect from the large vessels
· The authors report diameters also in their measurement (Table 1). How did they define the diameters here? Are they assuming that the tumor is spherical then the diameter is like an average radius of this sphere? The fact that there is a heat sink effect might affect the shape of the lesion too
· It is suggested to include more outcomes in the conclusions, especially from the quantitative point of view. Besides, it is also suggested to remark that using different procedures like saline infusion (doi.org/10.1016/j.apm.2016.11.032) or modulated heat might be a solution to dump the heat sink effect. All this should be of help for people that have to treat HCC in zones where the heat sink effect is really high
Author Response
Dear Ms. Aileen Han
and the reviewers, Cancers
Manuscript ID: cancers-2870456
" Usefulness of body position change during local ablation therapies for hepatocellular carcinoma "
Thank you for your kind review of our manuscript.
We appreciate receiving this opportunity to re-submit our manuscript.
Considering the suggestions, we have written Point by Points. We believe that these changes will alleviate the concerns of the reviewers. We would greatly appreciate if our manuscript could be accepted for publication in Cancers.
Sincerely,
Hitomi Takada, MD.
Gastroenterology and Hepatology Department of Internal Medicine, Faculty of Medicine,
University of Yamanashi,
Yamanashi, Japan
Point by points
- Reviewer 1
- When choosing the patients, how did the authors be sure that among their patients they have many of this with lesions close to large vessels? Did they somehow quantify the distance between a lesion and a large vessel, and use a threshold criterion for this?
Thank you very much for your detailed comments.
In this report, we used axial section CT images at 1-mm intervals, coronal section images at 3-mm intervals and sagittal section images at 3-mm intervals to establish the location of the lesion and the large vessel. We set the threshold for the distance between HCC and a large vessel at 5 mm, referring to previous reports.
- Teratani T, Yoshida H, Shiina S, Obi S, Sato S, Tateishi R, et al. Radiofrequency ablation for hepatocellular carcinoma in so-called high-risk Hepatology. 2006;43(5):1101-8.
We have modified the text as follows. (line 120, page 3)
HCC nodules were classified as HCC in high-risk locations if they were adjacent to large vessels or extrahepatic organs or if they were poorly visible on US. Nodules were considered adjacent to large vessels if they were located <5 mm from the first or second branch of the portal vein, the base of hepatic veins, or the inferior vena cava, referring to previous reports. Nodules located <5 mm from the diaphragm, heart, lung, gallbladder, right kidney, or gastrointestinal tract were considered adjacent to extrahepatic organs. The distance between the edge of the nodule and the large vessel or extrahepatic organ was measured using axial section CT images reconstructed at 1-mm intervals, coronal section images at 3-mm intervals and sagittal section images at 3-mm intervals.
- Did the authors account for the fact that if lesions are too close then the heat conducted within tissues would affect lesions? Is it fair to include patients with multiple lesions?
Thank you very much for your detailed comments.
In patients with multiple lesions, we also consider that heat conducted into the tissue during treatment of a target nodule may affect nodules in close proximity positions. In this study, no cases were included in which the distance between intrahepatic nodules was short, which may have reduced the effect on the results.
We have added the text as follows. (line 82, page 2)
In cases with multiple nodules, cases with two nodes in close proximity to each other, such that the predicted area of ablation at the time of ablation of target nodule included the other nodule, were excluded.
- Why did the authors use the Emprint ablation system? Is there an effect on ablation due to the internally cooled electrode, that might depend on the antenna employed?
Thank you very much for your comments.
This study only covers patients receiving RFA and does not include patients receiving microwave ablation therapies using Emprint. Due to errors in our description, we have corrected as follows. (line 107, page 3)
The devices used for RFA were the Cool-tip RFA system (Covidien, USA), and VIVA RF system (STARmed, Korea) (23).
The following statements in the original manuscript have been removed.
In the case of the Emprint electrode, we adjusted the radio wave output, as shown in the table, using probes ranging from short (15 cm) to standard (20 cm) and long (30 cm).
- The procedure used for ablation might affect results. The authors report in subsection 2.2 that they increase the output power up to 10 W/min until there is a too much high impedance. How did the authors monitor this?
Thank you very much for your comments.
As you pointed out, our description was inappropriate.
We have modified as follows. (line 110, page 3)
In the case of 3-cm internally cooled tip RF electrodes, the output was initiated at 60 W and increased by 20 W/min until tissue impedance overshooting occurred using the manual i mode and watching the output display on the generator.
- In this paper, the authors essentially study how a heat sink effect from large vessels might affect thermal ablation. If this is the aim of the present paper, it is suggested to also include in the state of art papers that present a throughout analysis of this effect (doi.org/10.3390/bioengineering10091057). This would underline why investigating this aspect is of primary importance in cancer ablation
Thank you very much for your detailed comments.
As you pointed out, we should have taken into account and stated the difference of a heat sink effect from large vessels in patients receiving RFA and MWA. We have modified the text as follows. (line 285, page 11, and line 361, page 13)
We have added the recommended references to our list.
It has been the most recent generation ablation system that uses electromagnetic waves that are less prone to the heat sink effect (34).
Fouth, the study only included patients who received RFA therapies and did not include those with MWA, which is widely used these days. There will be a difference in results between patients with RFA and MWA, especially when treating nodules near large vessels, and further studies are needed. Fifth,~
- When comparing outcomes from different patients, why didn't the authors give more emphasis to the input power as a variable? This has to compensate the heat sink effect from the large vessels
Thank you very much for your detailed comments.
As you pointed out, we should have considered output power as one of the related factors for the technical success rate. Although additional studies were conducted, there was no association between output power and success rate in this study. A possible reason for the lack of association between output power and success rate is that high-risk HCCs in this study vary widely in terms of adjacent to large vessels or extrahepatic organ and poorly visible on US, which may be related to the results. Further subdivisional studies are needed in the future. We have modified Table.2.
- The authors report diameters also in their measurement (Table 1). How did they define the diameters here? Are they assuming that the tumor is spherical then the diameter is like an average radius of this sphere? The fact that there is a heat sink effect might affect the shape of the lesion too
Thank you very much for your detailed comments.
In this study, we define 'Maximal diameter of the tumors' as the diameter at the largest plane obtained using axial section CT images at 1-mm intervals, coronal section images at 3-mm intervals and sagittal section images at 3-mm intervals. However, as you have pointed out, the shape of the tumor, as well as its diameter, may be associated with the effectiveness of therapies and prognosis.
We have added this issue to the discussion section as a future issue. (line 90, page 2, and line 358, page 13)
We defined 'diameter of the tumors' as the maximal diameter obtained using axial sec-tion CT images at 1-mm intervals, coronal section images at 3-mm intervals or sagittal section images at 3-mm intervals.
Third, there is a lack of detailed analysis of the shape of the tumor. Even nodules with the same maximum tumor diameter may differ significantly in their therapeutic efficacy and prognosis, depending on their three-dimensional structure.
- It is suggested to include more outcomes in the conclusions, especially from the quantitative point of view. Besides, it is also suggested to remark that using different procedures like saline infusion (doi.org/10.1016/j.apm.2016.11.032) or modulated heat might be a solution to dump the heat sink effect. All this should be of help for people that have to treat HCC in zones where the heat sink effect is really high
Thank you very much for your detailed comments.
As you pointed out, our description was too simple to convey the gist. Quantitative results have been added to the conclusion section, with reference to the literature you recommended.
We have modified as follows. (line 373, page 13)
To the best of our knowledge, this study is the first to statistically propose that BPC is safe and efficient in patients receiving local ablation therapies. The use of treat-ment-assist techniques such as BPC, artificial pleural fluid infusion, artificial ascites infusion, fusion imaging and CEUS, improved the intraoperative US visibility for high-risk HCC, compared to the baseline US. In particular, within the high-risk location group, nodules treated with BPC achieved a significantly higher technical success rate than those without BPC (99% vs. 91%, p = 0.015). BPC was the only factor related to the technical success rate for the high-risk location group (OR = 10, p = 0.034). In addition, no differences were found in the procedure time between the groups with and without BPC in the high-risk location group.
Reviewer 2 Report
Comments and Suggestions for Authors
In this study the authors present evidence of local ablation improvement in HCC patients depending on the patient position.
This is a well written technical article that provides usefull information for the clinicians involved in the HCC field.
Without wanting in any case to undersetimate the importance of the present study for the clinical doctor, I believe that this study should by published in a more technical and specialized journal.
Author Response
Dear Ms. Aileen Han
and the reviewers, Cancers
Manuscript ID: cancers-2870456
" Usefulness of body position change during local ablation therapies for hepatocellular carcinoma "
Thank you for your kind review of our manuscript.
We appreciate receiving this opportunity to re-submit our manuscript.
Considering the suggestions, we have written Point by Points. We believe that these changes will alleviate the concerns of the reviewers. We would greatly appreciate if our manuscript could be accepted for publication in Cancers.
Sincerely,
Hitomi Takada, MD.
Gastroenterology and Hepatology Department of Internal Medicine, Faculty of Medicine,
University of Yamanashi,
Yamanashi, Japan
Point by points
- Reviewer 2
Without wanting in any case to undersetimate the importance of the present study for the clinical doctor, I believe that this study should by published in a more technical and specialized journal.
Thank you very much for your kind comments.
Our current report may be biased towards technical details on local ablation therapies. Therapies for HCC are currently diverse, and there are not many hepatologists who are involved in ablation therapies among readers. However, as noted in the discussion section, the current era is changing, as local ablation therapies can be a treatment option not only for patients with early-stage HCC but also for those with advanced HCC as well. Therefore, we have thought this report should be shared with not only hepatologists who are involved in ablation therapies but chemotherapy specialists and physicians in other fields. We would greatly appreciate if our manuscript could be accepted for publication.
Reviewer 3 Report
Comments and Suggestions for Authors
The study examined the effectiveness of body position change during local ablation therapies for HCC. BPC significantly improved technical success rates in high-risk HCC locations (99% vs. 91%). However, BPC did not affect procedure time or long-term outcomes. Adjusting body position during ablation therapy for HCC in high-risk locations was safe and efficient. Some revisions are suggested.
1. The application of BPC needs to be thoroughly described, including its utilization for patients with various types of high-risk lesions and how it was incorporated during RFA procedures.
2. What factors should be considered when deciding to use BPC? Is it advantageous for patients with low-risk conditions?
3. Despite the higher success rate of RFA in high-risk lesions with BPC, the author should elucidate why high-risk patients with BPC show comparable recurrence and survival rates to patients without BPC.
4. It would be beneficial if the authors could compare whether BPC offers advantages over other treatment-assist methods, such as artificial pleural effusion, artificial ascites, CEUS, and RVS.
Comments on the Quality of English Language
Minor editing.
Author Response
Dear Ms. Aileen Han
and the reviewers, Cancers
Manuscript ID: cancers-2870456
" Usefulness of body position change during local ablation therapies for hepatocellular carcinoma "
Thank you for your kind review of our manuscript.
We appreciate receiving this opportunity to re-submit our manuscript.
Considering the suggestions, we have written Point by Points. We believe that these changes will alleviate the concerns of the reviewers. We would greatly appreciate if our manuscript could be accepted for publication in Cancers.
Sincerely,
Hitomi Takada, MD.
Gastroenterology and Hepatology Department of Internal Medicine, Faculty of Medicine,
University of Yamanashi,
Yamanashi, Japan
Point by points
- Reviewer 3
- The application of BPC needs to be thoroughly described, including its utilization for patients with various types of high-risk lesions and how it was incorporated during RFA procedures.
Thank you very much for your detailed comments.
As you pointed out, there was no specific information in our manuscript on the use of BPC or how to perform BPC during RFA procedures. We have illustrated the actual positions (Figure 2), and added procedures to the text. (line 143, page 4)
The numbers of figures have been corrected accordingly.
BPC was attempted not only in these locations, but also in many cases where we want to further improve the image characterization using US, where the nodule is classified as high-risk HCC and we need a safer puncture route, or where are expected to be difficult to puncture with the basic approach due to obesity and postoperative, etc. If BPC was judged to be effective when we performed planning US the day before treatment, all procedures started after appropriate positioning using a soft cushion, arm-board and support device. Treatment-assist techniques in planned combinations were performed as follows; artificial pleural fluid or ascites infusion was performed before BPC for puncture, and RVS and CEUS were performed just before puncture after BPC.
- What factors should be considered when deciding to use BPC? Is it advantageous for patients with low-risk conditions?
Thank you very much for your detailed comments.
As you pointed out, factors to consider when deciding to perform BPC and the significance of BPC in low-risk HCC group were not well described in our manuscript. Patients with no problems with image characterization of HCC using US and a safe puncture route were treated in the basic supine position. Contrastly, we believe that BPC should be attempted in all cases where there is a slight improvement in the image characterization using US, or where the nodule is classified as a high-risk HCC and a safer puncture route is to be sought. In addition, in patients with low-risk HCC, the technical success rates in the group with BPC were comparable to those in the group without BPC (100% vs. 94%, p=0.33). There was no difference in the procedure time, local tumor progression rates, intrahepatic distant recurrence rates, and OS between the two groups (p=0.16, 0.28, 0.12, 0.68).
We have modified as follows. (line 267, page 11, and line 306, page 12)
We believe that BPC should be attempted in all cases where we want to further improve the image characterization using US, or where the nodule is classified as high-risk HCC and a safer puncture route is to be sought.
In contrast, in patients with low-risk HCC, the technical success rates in the group with BPC were comparable to those in the group without BPC (100% vs. 94%, p=0.33). There was no difference in the procedure time, local tumor progression rates, intrahepatic distant recurrence rates, and OS between the two groups (p=0.16, 0.28, 0.12, 0.68). Therefore, it is important to have an adequate treatment plan for each nodule respectively before starting local ablation therapies and to ensure a safe and secure puncture route.
- Despite the higher success rate of RFA in high-risk lesions with BPC, the author should elucidate why high-risk patients with BPC show comparable recurrence and survival rates to patients without BPC.
Thank you very much for your detailed comments.
As you pointed out, we need to mention the divergence in the outcomes of success rates and those of prognosis when comparing with and without BPC. We believe that although the technical success rates in one session in the group with BPC were higher than in the group without BPC, there was no association between BPC and recurrence or survival rates because of the achievement of adequate ablation in multiple sessions and adequate HCC control with the addition of other therapies.
We have modified as follows. (line 304, page 12)
In addition, there was no association between BPC and recurrence or survival rates, probably because of the achievement of adequate ablation in multiple sessions, or ad-equate HCC control with the addition of other therapies.
- It would be beneficial if the authors could compare whether BPC offers advantages over other treatment-assist methods, such as artificial pleural effusion, artificial ascites, CEUS, and RVS.
Thank you very much for your detailed comments.
As you pointed out, it would be more useful if it could be shown that performing BPC has an impact on the technical success rates, procedure time and recurrence rates, and prognosis of high-risk HCC group compared to the other treatment-assist techniques. However, there are few cases in which each treatment-assist technique is used alone, and the appropriate technique varies from nodule to nodule. Therefore, it is difficult to prove the advantage of BPC over the other treatment-assist techniques only in this study. Further case accumulation is needed.
We have added to the Discussion section as follows. (line 365, page 13)
Moreover, there are few cases in which each treatment-assist technique is used alone, and the appropriate technique varies from nodule to nodule. Therefore, it is difficult to prove the advantage of BPC over the other treatment-assist techniques only in this study.
Reviewer 4 Report
Comments and Suggestions for Authors
The authors present on the the usefulness of body position change (BP9C during local ablation therapies in patients with HCC. In this study, 283 HCC nodules were included that underwent local ablation therapy.
High risk HCC criteria were defined by experienced investigators defined by the proximity to large vessels, positions adjacent to extrahepatic organs, or poor visibility under ultrasound (US) guidance.
The techniques were evaluated according tot he technical success rates, procedure time, and prognosis. 176 (62%) nodules were classified in the high-risk location.
The high-risk location group was treated with techniques such as BPC, artificial pleural fluid, artificial ascites, fusion imaging, and contrast-enhanced US more frequently than the low-risk location group. The technical success rates were 96% and 95% for the high- and low-risk location groups, respectively.
BPC emerged as the sole contributing factor to the technical success rate in the high-risk location group.
The results presented by the authors are sound and the hypothesis is clearly stated. However, the authors should interpret the results more thouroughly.
In the introduction, advancements in technology are discussed on page 2. Reference 14 includes navigation software. However, navigation is not mentioned in the text, and should be added as well in the introduction part as well in the disuccion section, where it is shortly mentioned, but not discussed into detail. The use of navigation software strongly influences treatment planning and therapeutic results of RFA and MWA, as it is an essential tool in needle placement techniques.
Reference 14: Minami T, Minami Y, Chishina H, Arizumi T, Takita M, Kitai S, et al. Combination guidance of contrast-enhanced US and fusion 384 imaging in radiofrequency ablation for hepatocellular carcinoma with poor conspicuity on contrast-enhanced US/fusion 385 imaging. Oncology. 2014;87 Suppl 1:55-62
In the Materials and Methods section, the authors do not elaborate on patient selection precisely. Are the included patients cirrhotic or non-cirrhotic patients? Was the Child Pugh Score assessed or not? Criteria as the BCLC criteria and EASL criteria mentioned in the text are only valid in cirrhotic patients.
As the authors discuss classical HCC findings, the reader would like to know whether LIRADS criteria were applied in HCC evaluation. Was late phase imaging performed during CT or MRI scans?
Regarding the intraprocedural techniques applied, it is not clear to the reader whether artificial ascites was applied befor or after BPC? How was the combination of techniques evluated? Did the applicated fluid shift position after BPC?
Regarding Table 1, is the diameter of the tumors mentioned (12 mm in all groups) the maximal, as written, or the median diameter?
In the Discussion section, Reference 2 includes the EASL Clinical Practice Guidelines: Management of hepatocellular carcinoma. J Hepatol. 2018;69(1):182-236. However, in the text the Japan Society of Hepatology is mentioned. Please explain!
Reference 2 is the EASL Clinical Practice Guidelines: Management of hepatocellular carcinoma. J Hepatol. 2018;69(1):182-236. However, in the text the Japan Society of Hepatology is mentioned.
Author Response
Dear Ms. Aileen Han
and the reviewers, Cancers
Manuscript ID: cancers-2870456
" Usefulness of body position change during local ablation therapies for hepatocellular carcinoma "
Thank you for your kind review of our manuscript.
We appreciate receiving this opportunity to re-submit our manuscript.
Considering the suggestions, we have written Point by Points. We believe that these changes will alleviate the concerns of the reviewers. We would greatly appreciate if our manuscript could be accepted for publication in Cancers.
Sincerely,
Hitomi Takada, MD.
Gastroenterology and Hepatology Department of Internal Medicine, Faculty of Medicine,
University of Yamanashi,
Yamanashi, Japan
Point by points
- Reviewer 4
- In the introduction, advancements in technology are discussed on page 2. Reference 14 includes navigation software. However, navigation is not mentioned in the text, and should be added as well in the introduction part as well in the disuccion section, where it is shortly mentioned, but not discussed into detail. The use of navigation software strongly influences treatment planning and therapeutic results of RFA and MWA, as it is an essential tool in needle placement techniques.
Reference 14: Minami T, Minami Y, Chishina H, Arizumi T, Takita M, Kitai S, et al. Combination guidance of contrast-enhanced US and fusion 384 imaging in radiofrequency ablation for hepatocellular carcinoma with poor conspicuity on contrast-enhanced US/fusion 385 imaging. Oncology. 2014;87 Suppl 1:55-62.
Thank you very much for your kind comments.
As you pointed out, RVS is an important method that has been noted to be associated with success rates among other modalities, and details should be added to this manuscript.
We have added description as follows. (line 325, page 12 and line 330, page 12)
In this study, there was no improvement in the success rates of the high-risk location group with these other modalities, although improved intraoperative US visibility for high-risk HCC. It is particularly interesting that the combined RVS was not a factor associated with increased success rates.
RVS was most frequently used in these cases, including those in which RVS was not used effectively. Therefore, it is not possible to conclude from this study whether they were useful when other modalities such as RVS were used in a planned manner, and further study is needed.
- In the Materials and Methods section, the authors do not elaborate on patient selection precisely. Are the included patients cirrhotic or non-cirrhotic patients? Was the Child Pugh Score assessed or not? Criteria as the BCLC criteria and EASL criteria mentioned in the text are only valid in cirrhotic patients.
Thank you very much for your kind comments.
As you pointed out, there was a lot of description about the target nodule and not enough about the target patient. We have modified as follows.
- We have revised the description in the Table.1 for whether the patient has cirrhosis or not, using a value of platelets < 150,000, i.e., a value that may be associated with fibrosis of F2 or higher in patients with viral hepatitis or suspicious for cirrhosis in patients with NASH.
- For Child Pugh grade, the description in Table.1 was changed to use Child Pugh score.
- The BCLC criteria are described in the Discussion section when presenting general frequencies, etc., but are not used in presenting the results of this study, therefore, no specific changes have been made to the text.
- As the authors discuss classical HCC findings, the reader would like to know whether LIRADS criteria were applied in HCC evaluation. Was late phase imaging performed during CT or MRI scans?
Thank you very much for your kind comments.
As you pointed out, the lack of description using LI-RADS criteria made the description difficult to understand the content. In this study, all target nodules at baseline exhibited LR-4 or 5. Furthermore, nodules that achieved 'LR-TR nonviable' on late CT/MRI evaluation were defined as 'technical success'.
We have added description using LI-RADS criteria as follows. (line 85, page 2 and line 164, page 4)
Nodules that showed pathological examination result, or non-rim hyperenhancement in the arterial phase of dynamic computed tomography (CT) or gadolinium ethoxybenzyl diethylenetriamine penta-acetic acid-contrast-enhanced magnetic resonance imaging (MRI) and non-peripheral washout or threshold growth, that is, only nodules that showed LR-4, 5 using LI-RADS (Liver Imaging Reporting and Data System) were diagnosed as HCC 22.
Nodules showing no CT/MRI evidence of residual tumor with continuous monitoring <3 months, i.e., nodules that achieved ‘LR-TR nonviable’ were defined as 'technical success' 22.
- Regarding the intraprocedural techniques applied, it is not clear to the reader whether artificial ascites was applied befor or after BPC? How was the combination of techniques evluated? Did the applicated fluid shift position after BPC?
Thank you very much for your kind comments.
As you pointed out, the procedure for combining treatment-assist techniques was not adequately described. In particular, artificial pleural fluid or ascites infusion was performed before BPC for puncture, and although the migration of the injected fluid was confirmed in many cases, there were some cases in which no migration was observed due to postoperative effects.
We have added description as follows. (line 149, page 4)
Treatment-assist techniques in planned combinations were performed as follows; artificial pleural fluid or ascites infusion was performed before BPC for puncture, and RVS and CEUS were performed just before puncture after BPC.
- Regarding Table 1, is the diameter of the tumors mentioned (12 mm in all groups) the maximal, as written, or the median diameter?
Thank you very much for your detailed comments.
As you pointed out, there was our description of the diameter of the tumors was inadequate. We used maximum diameter as the "diameter" in this study, and defined 'Maximal diameter of the tumors' as the diameter at the largest plane obtained using axial section CT images at 1-mm intervals, coronal section images at 3-mm intervals and sagittal section images at 3-mm intervals.
We have modified description as follows. (line 90, page 3)
We defined 'diameter of the tumors' as the maximal diameter obtained using axial section CT images at 1-mm intervals, coronal section images at 3-mm intervals or sagittal section images at 3-mm intervals.
- In the Discussion section, Reference 2 includes the EASL Clinical Practice Guidelines: Management of hepatocellular carcinoma. J Hepatol. 2018;69(1):182-236. However, in the text the Japan Society of Hepatology is mentioned. Please explain! Reference 2 is the EASL Clinical Practice Guidelines: Management of hepatocellular carcinoma. J Hepatol. 2018;69(1):182-236. However, in the text the Japan Society of Hepatology is mentioned.
Thank you very much for your detailed comments.
As you pointed out, incorrect references were listed. We have modified description as follows.
(33) Kudo, M.; Kawamura, Y.; Hasegawa, K.; Tateishi, R.; Kariyama, K.; Shiina, S.; Toyoda, H.; Imai, Y.; Hiraoka, A.; Ikeda, M.; et al. Management of Hepatocellular Carcinoma in Japan: JSH Consensus Statements and Recommendations 2021 Update. Liver Cancer 2021, 10 (3), 181-223. DOI: 10.1159/000514174 From NLM.
Round 2
Reviewer 1 Report
Comments and Suggestions for Authors
The paper can be accepted as it is in the revised form
Reviewer 2 Report
Comments and Suggestions for Authors
Despite the significant improvement of the manuscript, I still believe that the present study is more suitable to a more technical and specialized journal.
Reviewer 3 Report
Comments and Suggestions for Authors
My questions have been addressed.